# Optimizing 3D Food Printing of Surimi via Regression Analysis: Physical Properties and Additive Formulations

**DOI:** 10.3390/foods14050889

**Published:** 2025-03-05

**Authors:** Jong Bong Lee, Na Young Yoon, Yeon Joo Bae, Ga Yeon Kwon, Suk Kyung Sohn, Hyo Rim Lee, Hyeong Jun Kim, Min Jae Kim, Ha Eun Park, Kil Bo Shim

**Affiliations:** 1Department of Food Science and Technology, Pukyong National University, 45 Yongso-ro, Nam-Gu, Busan 48513, Republic of Korea; whdqhd11@pukyong.ac.kr (J.B.L.); yeonjoo4459@gmail.com (Y.J.B.); gayeon7401@gmail.com (G.Y.K.); jhshon2@naver.com (S.K.S.); lhr2008@pukyong.ac.kr (H.R.L.); hyeongjun5767@gmail.com (H.J.K.); rlaalswo4992@naver.com (M.J.K.); phe5765@naver.com (H.E.P.); 2Food Safety and Processing Research Division, National Institute of Fisheries Science, Busan 46083, Republic of Korea; dbssud@korea.kr

**Keywords:** surimi, 3D food printer, rheological properties, surimi ink, additive

## Abstract

This study aimed to optimize the three-dimensional (3D) printing parameters for surimi-based inks and investigate the effects of additives (starch, salt, and water) on the rheological and textural properties of surimi paste, aiming to develop a universal formulation applicable across three fish species: Alaska pollock, golden threadfin bream, and hairtail. By analyzing the hardness, adhesiveness, storage modulus (G′), and complex viscosity of the surimi inks, a formula was developed to identify the range of physical properties required for stable and precise 3D printing. The parameter windows to build a 3D structure with a 45° slope were as follows: hardness, 150–415 g/cm^2^, and adhesion, −300 to −115 g. Mixing surimi with additives such as water, salt, and starch to obtain the desired physical properties facilitated the printing of 3D surimi samples using a 3D food printer.

## 1. Introduction

The three-dimensional (3D) printing of food is a promising technique for forming food products that meet consumer preferences regarding shape, texture, and nutrient content. Therefore, it is drawing increasing attention in the food, medical, pharmaceutical, and other fields [1,2]. In particular, 3D food-printing techniques have been studied to improve the edibility and properties of foodstuffs, as well as optimize the processing conditions. To prepare food-based “inks” for 3D printing, several foods have been combined with hydrocolloids [3,4,5] and highly viscous emulsion- and hydrocolloid-based materials that can be easily blended with various compounds, such as proteins, fats, prebiotics, probiotics, vitamins, and minerals, which have also been investigated [6,7,8]. The 3D printing of food inks provides high flexibility because of the nonstructured shape of the output. However, the food ink must have appropriate physical properties to ensure stable printing, layer stacking, and a robust structure. Of note, during the extrusion phase, fluidity is required, which can be achieved through plasticization, as seen in chocolate, syrups, and viscous hydrocolloids [9]. In contrast, during the stacking phase, the food ink should be cohesive [10]. The cohesivity of an ink is related to its mechanical properties, such as viscosity, storage modulus (G′), and hardness [11]. Such properties can be modified using additives, for example, peanuts with xanthan gum, mushrooms with starch, and plant-based meats with hydrocolloids [12,13,14,15]. However, the salt-soluble proteins in meat and fish-based animal products form characteristic plasticity [16]. In particular, animal-based protein can be adjusted by adding hydrocolloid binders and lipids [17].

Surimi is a seafood product made from fish with added sugars, salts, starches, and alcohols, yielding a gel comprising concentrated protein myofibrils [18,19]. The plasticity, high viscosity, and cohesiveness of surimi paste, resulting from the salt-soluble proteins actin and myosin, make it suitable as a 3D printing ink. Additionally, surimi has high structural integrity because it undergoes thermal gelation. Therefore, surimi remains stable in cooking after extrusion [20]. However, the rheological and physical properties of surimi paste vary depending on fish species, additive content, and processing conditions, including the temperature and washing treatment [21], the latter being crucial [22,23]. In addition, the physical properties of surimi paste can be controlled by varying the additives added during processing. However, the formulation conditions and additive contents must be studied to ensure proper physical properties, including plasticity and cohesiveness, but, to date, studies have been limited to shrimp and chicken [18,24]. Furthermore, research on using surimi ink for 3D printing has been limited to simple layer printing [25,26]. Exploring printing methods that eliminate the need for support structures when using surimi inks is essential to enhance consumer customization and increase its scalability for general use.

This study aimed to develop a universal method for determining the appropriate mixing conditions of surimi inks based on 3D printing characteristics and consumer preferences, regardless of the fish species and its characteristics. In particular, a set of parameter windows necessary for printing surimi with a 45° slope without any support was identified. Central composite design (CCD), an optimization technique based on regression analysis, was used to optimize multiple parameters simultaneously, as well as examine their influence on the ink properties. CCD was chosen because of its ability to investigate linear and quadratic models based on a relatively large number of design points, reflecting the curvature of the regression model, making it highly suitable for model optimization [27]. As food matrices, we selected three fish species whose surimi forms have different physical strengths (Alaska pollock (*Gadus chalcogrammus*), golden threadfin bream (*Nemipterus virgatus*), and hairtail (*Trichiurus haumela*)). The corresponding model was used to develop universal physical properties for the 3D printing of surimi, regardless of fish species. An optimal range and contents of additives (starch, salt, and water) were established for stable 3D printing using the obtained regression model. The results of this study will increase the efficiency of 3D food printing and enhance the development of foods tailored to consumer preferences.

## 2. Materials and Methods

### 2.1. Materials

The three types of surimi (Alaska pollock (*G. chalcogrammus*, FA grade, USA), golden threadfin bream (*N. virgatus*, FA grade, Thailand), hairtail (*T. haumela*, FA grade, China)) were purchased from a local company (ASL Trading Co., Ltd., Busan, Korea). The frozen surimi was cut into blocks (approximately 1 kg), sealed in a vacuum bag, and maintained at −18 °C for approximately one month. Before use, the frozen surimi was partially thawed at room temperature for 1 h to allow the core temperature to reach approximately −5 °C [28]. Salt and starch were purchased from local shops.

### 2.2. Preparation of Surimi Paste

Each frozen surimi sample was thawed at 4 °C for 2 h and cut into cubes (2 × 2 × 2 cm). Then, the cubed surimi (1 kg) was chopped using a food processor (UMX-5; Stephan Machinery GmbH, Hameln, Germany) at 1500 rpm for 1 min. Subsequently, salt and starch were added to the food processor, and the surimi was blended at 1500 rpm for 2 min. Finally, water was added to the food processor and blended at 1500 rpm until the sample temperature reached a temperature of 7.5 °C. The starch, water, and salt contents were determined according to the CCD conditions. The food processor was maintained under vacuum at 4 °C to avoid the formation of bubbles in the surimi ink.

### 2.3. Screw-Based 3D Food Printer and 3D Model

The 3D food printer used in this study was a screw-type printer that extrudes food ink by applying pressure to a syringe nozzle (volume: 100 mL, nozzle diameter: 1.94 mm) using a step motor. A 3D food printer was created using Makerbox (3D PrintingNMakerbox Co., Ltd., Seoul, Korea) (Figure 1). A 3D model (bottom length: 15 mm; height: 15 mm; slope: 45°) was designed using Fusion (v.2.0.19941, Autodesk, San Francisco, CA, USA) and sliced using Cura (ver. 5.2., Ultimaker, Utrecht, The Netherlands) according to the printing parameters (Table 1). The extrusion speed was 5 mm/s (15 mm × 30% printing speed × flow rate), as determined through trial tests. The shear rate of the surimi ink was calculated by multiplying the actual printing speed by the volume, as shown in Equation (1).(1)$$D_{v}=l\times \frac{S}{200\;step}$$
where *l* is the length of the screw lead (8 mm), and *S* is the number of steps per distance (10 steps/mm). The theoretical volumetric flow rate was 5.91 mm^3^/s, and the actual volumetric flow rate was 5.75 mm^3^/s. The shear rate ($\dot{\gamma }$) of the food ink extruded from the nozzle, as given by Equation (2), is the ratio of the volumetric flow rate of the ink (*Q*) to the thickness of the slot (*b*) and the distance between the nozzle and the printing stage (*h*) [29]. The shear rate applied to the extrusion of the surimi ink was approximately 1 rad/s. All printing parameters are listed in Table 1.(2)$$\dot{\gamma }=\frac{6Q}{{bh}_{2}}$$

### 2.4. Printability

The criteria for assessing the quality of the extruded surimi included the slope angle (the maximum sustainable angle without collapse) and layer sharpness, as measured using ImageJ (ver. 1.54d, National Institutes of Health, Bethesda, MD, USA). The slope angle was measured using the outer angles of the left and right sides of the printed surimi from the front. The sharpness of the printed surimi was measured by setting a region of interest (ROI) of 300 × 50 pixels at the top and calculating the ratio of the peak intensity threshold, i.e., the contrast. The criteria for the quality assessment of the printed surimi are listed in Table 2.

### 2.5. Physical Properties of Surimi

#### 2.5.1. Rheological Properties

The rheological properties of the surimi ink were measured using a rheometer (DHR2, TA Instruments, New Castle, DE, USA) equipped with an aluminum parallel plate (diameter: 25 mm). Two experiments were performed: (1) oscillation amplitude and (2) oscillation frequency sweeps. Each experiment was repeated six times, and the data were averaged for analysis. The distance between the probe and the plate was 1 mm, and the temperature was maintained at 25 °C during measurements.

##### Oscillation Amplitude Tests

The storage modulus (G′), loss modulus (G″), and loss tangent (tan(*δ*) = G″/G′) were measured using the oscillation amplitude tests. The tests were performed at a strain range of 0.0125–12.5% and a constant angular frequency of 10.0 rad/s. The linear viscoelastic region (LVR) of the surimi ink was measured for the oscillation frequency test.

##### Oscillation Frequency Tests

The complex viscosity was measured using oscillation frequency tests. The tests were performed at an angular frequency range of 0.1–100 rad/s, and the strain (%) was set to the limit point of the linear viscoelastic region determined in the oscillation amplitude tests.

#### 2.5.2. Texture Profile Analysis

Texture profile analysis (TPA) was performed as described by Rodríguez-Herrera et al. [30] with some modifications. Surimi paste was placed in a cylindrical vessel (40 × 60 mm). The filled vessel was placed in a texture profile analyzer (CR-100, Sun Scientific Co., Ltd., Tokyo, Japan) equipped with a 10 kg load cell and 20 mm diameter cylindrical probes. Samples were compressed twice at a crosshead speed of 60 mm/min to 60% of the sample height. The hardness, adhesiveness, cohesiveness, and springiness were measured.

### 2.6. Central Composite Design for Regression Analysis

A regression model was designed using the CCD module in Minitab v. 19 (Minitab Inc., State College, PA, USA) to study the effects of the additive contents (starch, water, and salt) on the physical properties of the ink.

The independent variables were starch (*x*_1_), water (*x*_2_), and salt (*x*_3_) content, and the values were determined step by step based on commercial additions to make the surimi-based products [31] (Table 3). The dependent variables were hardness (*y*_1_), adhesiveness (*y*_2_), storage modulus (*y*_3_), and complex viscosity (*y*_4_).

### 2.7. Statistical Analysis

The physical properties of samples were analyzed using four different methods: (1) analysis of variance (ANOVA), (2) Tukey’s multiple comparison test, (3) correlation analysis, and (4) a paired *t*-test using SPSS (version 27, SPSS Inc., Chicago, IL, USA). Results were considered statistically significant at *p* < 0.05.

## 3. Results and Discussion

### 3.1. Printability

Three types of surimi were extruded according to the CCD conditions, and the results of the quality assessment are presented in Table 4. The extruded surimi was categorized into three groups (Figure 1). Group A (Figure 1) showed the most stable printability. The slope angle and sharpness were 46.06 ± 10.7° and 39.83 ± 7.21%, respectively. Additionally, after extrusion, Group A surimi maintained its structure for some time after printing. Group B (Figure 1) was unstable compared to Group A. The slope angle of Group B (56.31 ± 6.39°) also suggested instability. However, the layer sharpness was excellent, having a sharpness of 39.58 ± 7.54%. Group C (Figure 1) showed high deformation and an unstable structure (slope angle, 68.90 ± 8.30°; sharpness, 55.80 ± 7.57%). Alaska pollock under CCD condition no. 11 showed noncontinuous extrusion. Therefore, no. 11 of the Alaska pollock was considered unsuitable for 3D printing and assigned to Group C.

### 3.2. Correlation Between Physical Properties

Correlation analysis was conducted using Pearson’s correlation coefficient between the physical properties and printability of each surimi ink. The results are shown in Figure 2. For golden threadfin bream, factors such as hardness, G′, G″, and complex viscosity exhibited a significant positive correlation, ranging from 0.809 to 0.864. In contrast, cohesion and adhesion showed a negative correlation, with values between −0.498 and −0.693.

Alaska pollock surimi had a positive correlation with hardness, G′, and complex viscosity, with correlation coefficients ranging from 0.571 to 0.704. In contrast, it displayed a negative correlation with adhesion, with a correlation coefficient of −0.633. For hairtail surimi, only hardness and adhesiveness were measured, showing correlation coefficients of 0.710 and −0.713, respectively. This pattern was consistent with previous studies on white croaker (*Pennahia argentata*) and white-leg shrimp (*Litopenaeus vannamei*) [18,32]. Combining the findings from previous studies with the results for the three types of surimi studied herein revealed that the effect of additives such as starch on the extrusion performance and printability of surimi ink was mainly determined by factors like hardness, fluidity (G′ and G″), and viscosity.

However, in the case of hairtail surimi, the correlation with other fish species was found to be very low. Therefore, it was important to select an evaluation index that can be applied to multiple types of surimi simultaneously. The low correlation coefficients for hairtail surimi were attributed to the differences in myofiber content among different fish species. Hairtail surimi has low myofibril content, which is only 7% among white meat fish species [33]. As a result, even when the same additives are added, a weaker gel network forms due to the low myofibril content, which leads to low viscosity of the surimi paste and gelling agent during cooking [34].

Variations in gel fluidity due to differing myofibrillar contents among fish species complicate the examination of additives properties [35]. Therefore, the evaluation of the performance indicators of surimi ink had to be met to prioritize printing safety [30,36]. The flow properties involved in the process of extruding surimi are crucial indicators of ink performance. Notably, a tan(*δ*) value (the ratio of G″ to G′) of less than 1 indicates elastic behavior and relatively low fluidity. However, gels with low flow characteristics could still exhibit plastic flow when subjected to high mechanical strength. During the 3D printing process, high hardness and adhesion were essential for stably stacking layers and maintaining the stability of the printed model.

To identify the optimal range of physical properties for surimi, this study aimed to define more intuitive conditions using two key factors: hardness and adhesiveness. Hardness refers to the ability of the model to withstand the load as further ink is deposited, whereas adhesion indicates the ability to form a slope and enhances the model’s rigidity.

### 3.3. Rheology and Texture Profile

The results of the physical properties of surimi inks with different additive contents are plotted in Figure 3. Specifically, Figure 3a–c compare the physical properties of samples with different amounts of starch, water, and salt, respectively. The frequency sweep analysis shows how the deformation characteristics change with varying frequencies applied to the surimi. At low frequencies (around 0.1 rad/s), the material exhibited sufficient response time, causing the loose gel network to behave like a fluid. In contrast, at high frequencies (approximately 100 rad/s), the elastic response of the material was enhanced [16]. All surimi samples showed an increase in both G′ and G″ as the applied frequency increased. However, at the 1 rad/s condition used for the surimi extrusion in this study, the G′ values of all surimi samples were greater than the G″ values. This indicates that surimi exhibits plastic flow behavior during the extrusion process.

The G′ value of surimi can be influenced by additives, particularly starch, salt, and water. A higher starch content, combined with lower salt and water contents, leads to a higher G′ value, which indicates the formation of a harder gel. Starch absorbs moisture without competing with the proteins present in surimi, resulting in a “packing effect” that enhances the rigidity of the gel matrix [37]. Conversely, when salt is added, it dissolves and breaks down the salt-soluble proteins in the surimi. This process causes electrostatic shielding, which weakens the bonds between myofibrillar proteins and increases their fluidity [38]. Water increases the distance between bonds in the myofibrillar proteins, weakening electrostatic interactions and reducing the mechanical strength [16,39].

These effects of the different amounts and types of additives aligned with previous findings [17,39,40]. Viscosity reflects the changes in friction during the extrusion process of surimi ink during 3D printing. Regardless of the type of additives used, the viscosity decreases as the frequency increases, indicating that all three types of surimi demonstrate shear-thinning behavior. The changes in viscosity based on the content of additives were consistent with those observed for G′ and G″ [41]. Furthermore, the viscosity characteristics are indicative of the extrusion printing potential of surimi ink.

As the shear rate increased, the composite viscosity decreased in the range of 2000 to 18,000 Pa∙s and was below 3000 Pa∙s at 1 rad/s, where the extrusion of surimi was carried out. This result indicates that surimi ink is capable of stable extrusion within this range [40].

The results of the texture profile analysis were consistent with those of the rheology analysis. Alaska pollock surimi exhibited high levels of hardness and adhesiveness, whereas hairtail surimi showed the lowest levels of both. The texture of surimi varied significantly based on the amount of moisture added (Figure 3b), whereas the addition of starch had a minimal effect (Figure 3a).

Starch enhances the physical properties of surimi; however, moisture plays a more significant role in determining these properties. This is due to the involvement of water in the gelation and swelling processes of proteins [41]. According to the CCD results (Table 4), stable output conditions were mainly observed for Alaska pollock surimi, except for run No. 11, which included some golden threadfin bream and hairtail surimi. Consequently, the range of physical properties for stable surimi output was established based on the results of the texture profile analysis.

### 3.4. Regression Model of Physical Properties

The analysis of CCD conditions for hardness, adhesiveness, G′, and complex viscosity, which are closely tied to printability, is summarized in Table 5. The regression coefficients and model performance for surimi independent variables are shown in Table 6.

The coefficients of determination (*R*^2^) for hardness (*y*_1_) and adhesiveness (*y*_2_) were high across all samples: 96.48–99.37% and 93.40–99.35%, respectively.

For Alaska pollock surimi, G′ (*y*_3_) and complex viscosity (*y*_4_) also had high *R*^2^ values of 99.33% and 99.59%, respectively. However, golden threadfin bream surimi showed lower correlations for G′ (89.31%) and complex viscosity (89.53%), and hairtail surimi had the lowest *R*^2^ values for these properties at 48.08% and 47.83%, respectively. Statistically, *R*^2^ values above 70% are significant [42].

The regression model for starch (*x*_1_) yielded *p*-values ranging from 0.21 to 0.93, indicating poor predictive performance. Although the starch addition increased hardness and stiffness (e.g., G′) by cross-linking with proteins [43], excessive starch weakened these cross-links [44], reducing the predictive power of the model. In contrast, the models for water (*x*_2_) and salt (*x*_3_) were significant. For moisture, the *p*-values were highly significant (*p* < 0.001 for Alaska pollock; *p* < 0.05 for golden threadfin bream and hairtail).

Overall, the physical properties of surimi are strongly influenced by the water (*x*_2_) and salt (*x*_3_) content, making them effective predictors. The polynomial regression model for hardness and adhesiveness, which correlated highly across all surimi types, is the most suitable for determining the optimal additive content for stable 3D printing.

### 3.5. Optimal Additive Range

The sample classifications based on physical properties and printed samples prepared under different CCD conditions are shown in Figure 4. The ranges of hardness and adhesiveness values for stable printing were 150–415 g/cm^2^ and −300 to −115 g, respectively. In this study, the range of additives for the three surimi samples used for 3D printing was calculated using a polynomial regression model based on hardness and adhesiveness (Table 5). Therefore, the intersection of the pairs of additive contents that simultaneously satisfy the six regression equations (three types of surimi and two physical properties: hardness and adhesiveness) can be considered as the range for stable 3D printing, regardless of the type of surimi. These pairs of additive contents are sets of continuous values. Figure 5 shows an example of an additive content combination for the stable 3D food printing of the three species. Within these windows, even if surimi from different fish species is used, the additive content (starch, water, salt) can be complementarily substituted for, provided that the physical properties of the surimi ink are similar [18].

### 3.6. Deviations in Physical Properties Within Optimal Additive Range

Finally, we evaluated the printing properties of the proposed range of additive contents. The combinations of surimi additives for the significance tests were randomly selected, and the predicted and observed values for each condition are listed in Table 7. The prediction errors for hardness and adhesiveness (respectively) were as follows: Alaska pollock, 96.7–107.2% and 96.9–105.3%; golden threadfin bream, 75.9–110.8% and 72.7–119.8%; and hairtail, 96.2–109.4% and 95.9–107.9%. The *p*-value of the *t*-test between actual and predicted values for each sample was 0.111–1.000, indicating no significant differences for each property (*p* > 0.05). Therefore, it is possible to predict the quantitative variation in the properties using the optimized additive ranges.

## 4. Conclusions

The two main factors determining the suitability of surimi ink for 3D printing are intrinsic factors (gelling agents in each fish species) and extrinsic factors (additives). In this study, we used regression analysis with central composite design to investigate the effect of different surimi ink compositions on their physical properties. The addition of starch enhanced the physical strength (i.e., hardness, G′, and G″), whereas water and salt enhanced the fluidity (i.e., viscosity and adhesiveness). When the additive contents were adjusted to obtain a hardness of 150–415 g/cm^2^ and adhesiveness of −300 to −115 g, stable printing, achieving a 45 °C slope, was achieved. Therefore, the properties of surimi could be adjusted for 3D printing regardless of fish species. Finally, we identified the optimal range of additives that simultaneously satisfy six regression models (three types of surimi and two 3D printing properties). Crucially, the findings of this study indicate that the suitability of 3D food-printing inks can be determined before printing by analyzing key physical properties such as hardness and adhesiveness or by varying additive contents.

This study suggested an optimal formula for surimi 3D printing ink, regardless of fish species. Stable output could be achieved by following specific ranges of physical properties, regardless of the additives and fish species used. This scalability allows for the creation of formulations that incorporate functional or nutritional additives, as well as ingredients that enhance flavor using surimi as 3D printing ink.

## Figures and Tables

**Figure 1 foods-14-00889-f001:**
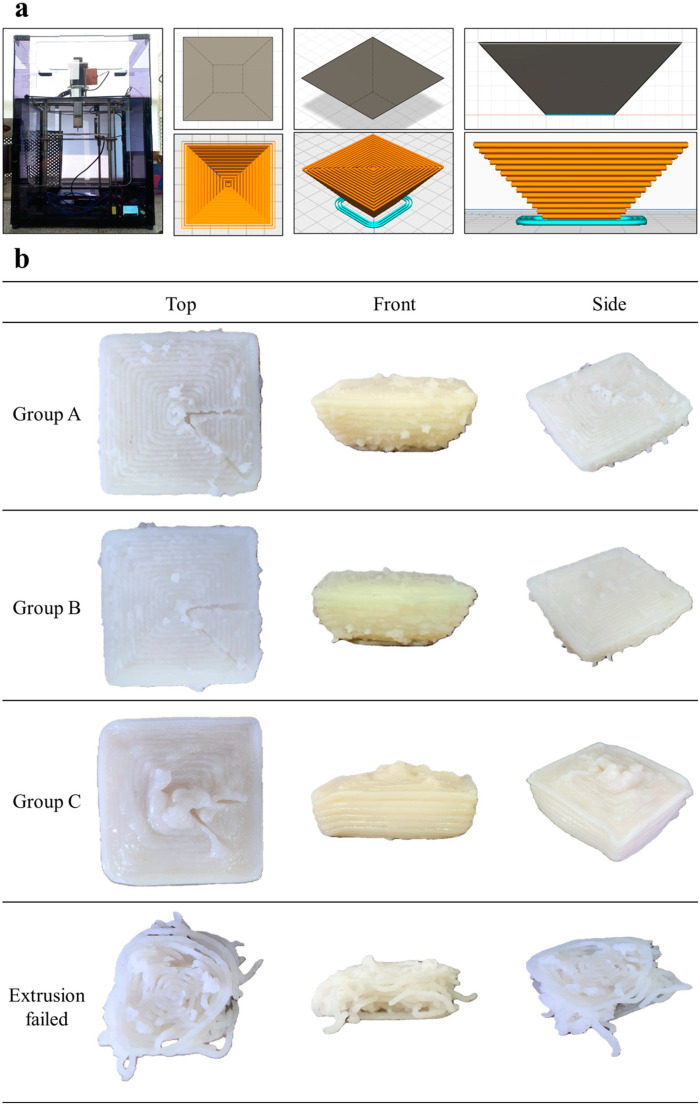
Photographs of 3D modeling and printed surimi: (**a**) 3D food printer, 3D model, and (**b**) example of 3D-printed foods with different stabilities.

**Figure 2 foods-14-00889-f002:**
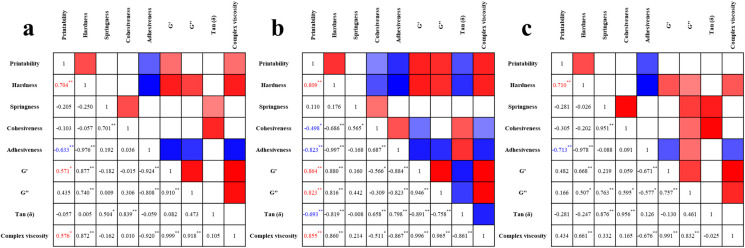
Correlation between printability and physical properties of 3D printing ink: (**a**) Alaska pollock, (**b**) golden threadfin bream, and (**c**) hairtail surimi. Two color scales are used: a red scale that covers positive correlation, and a blue scale covers negative correlation. * Correlation is significant at the 0.05 level (2-tailed). ** Correlation is significant at the 0.01 level (2-tailed).

**Figure 3 foods-14-00889-f003:**
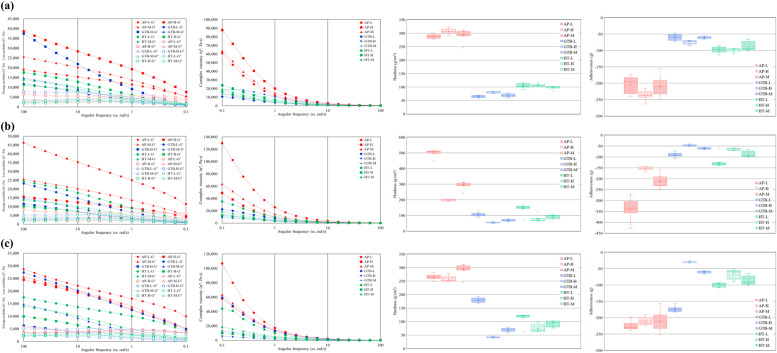
Rheology and texture profile analysis of prepared surimi inks: (**a**) comparison of physical properties of three types of surimi according to starch content; (**b**) comparison of physical properties of three types of surimi according to water content; and (**c**) comparison of physical properties of three types of surimi according to salt content.

**Figure 4 foods-14-00889-f004:**
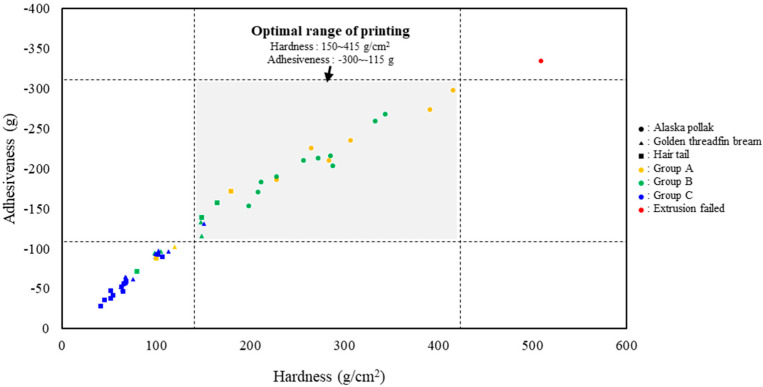
Optimal range for printing based on the physical properties of surimi ink.

**Figure 5 foods-14-00889-f005:**
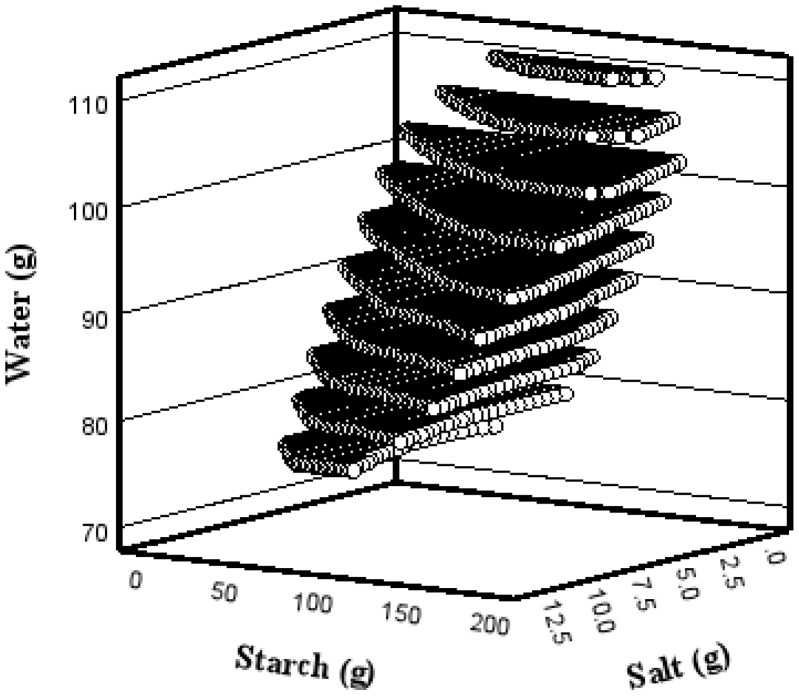
Example of surimi additive composition derived from the regression model.

**Table 1 foods-14-00889-t001:** Printing parameters of 3D food printer.

Parameter	Condition
Print speed (stage movement speed) (mm/s)	15
Flow rate (%)	0.3
Nozzle inner diameter (mm)	1.94
Initial nozzle height (mm)	1.20
Infill pattern	Concentric
Infill density (%)	80
Number of outer shells	0
Temperature (°C)	25

**Table 2 foods-14-00889-t002:** Quality criteria for 3D-printed surimi.

Parameter	Condition	Score
Angle of slope	Within 10% deviation	+1
	Over 10% deviation	0
Peak intensity threshold	Within 50% dark area	+1
	Over 50% dark area	0

**Table 3 foods-14-00889-t003:** Coded values of factor and level used for CCD.

Independent Variables	Code	−α	−1	0	+1	+α
Starch (g)	*x* _1_	10	45	100	155	190
Water (g)	*x* _2_	10	85	200	315	390
Salt (g)	*x* _3_	1	10	25	40	50

**Table 4 foods-14-00889-t004:** Printability assessment and sample grouping.

Run No.	Alaska Pollock			Golden Threadfin Bream			Hairtail		
	Slope Angle	Dark Area	Group	Slope Angle	Dark Area	Group	Slope Angle	Dark Area	Group
1	47.70 ± 3.01	41.58 ± 0.65	A	54.39 ± 0.41	35.57 ± 0.06	B	52.51 ± 7.81	39.45 ± 0.41	B
2	43.42 ± 4.17	44.82 ± 2.02	A	56.89 ± 6.92	48.92 ± 8.33	B	56.37 ± 1.63	39.78 ± 1.26	B
3	51.96 ± 0.68	38.06 ± 3.18	B	75.42 ± 10.74	55.48 ± 4.52	C	60.96 ± 2.48	55.22 ± 6.15	C
4	51.94 ± 4.02	38.43 ± 8.09	B	55.68 ± 6.58	44.88 ± 3.95	C	65.32 ± 8.29	61.11 ± 1.31	C
5	54.94 ± 2.73	35.68 ± 6.20	B	79.76 ± 1.99	65.62 ± 0.51	C	60.76 ± 3.59	45.81 ± 0.63	C
6	52.99 ± 3.45	29.97 ± 0.07	B	69.34 ± 1.77	63.07 ± 7.69	C	63.91 ± 1.35	44.08 ± 2.70	C
7	66.94 ± 2.91	46.15 ± 1.29	B	80.88 ± 4.53	61.39 ± 4.95	C	68.74 ± 3.48	57.41 ± 3.86	C
8	49.92 ± 3.61	32.05 ± 1.07	A	75.83 ± 3.21	56.58 ± 7.80	C	68.08 ± 5.56	65.99 ± 0.08	C
9	50.26 ± 6.20	39.86 ± 1.71	B	75.46 ± 2.50	56.89 ± 1.70	C	65.15 ± 2.62	54.98 ± 0.53	C
10	47.55 ± 2.89	31.89 ± 6.62	A	64.20 ± 0.96	46.52 ± 4.98	B	64.30 ± 0.23	36.28 ± 2.09	B
11	Extrusion failed		C	35.67 ± 27.59	45.95 ± 3.44	B	68.77 ± 2.05	50.78 ± 2.71	B
12	57.16 ± 1.96	49.94 ± 1.59	B	80.98 ± 2.59	73.52 ± 3.16	A	73.07 ± 1.51	52.74 ± 2.32	C
13	46.62 ± 4.24	48.54 ± 2.58	A	45.38 ± 0.09	36.41 ± 5.94	C	49.52 ± 2.25	44.50 ± 1.38	A
14	51.81 ± 1.16	31.62 ± 3.74	B	77.76 ± 8.25	54.05 ± 0.05	A	61.64 ± 4.11	51.90 ± 3.33	C
15	55.82 ± 2.69	35.73 ± 1.04	B	67.02 ± 8.53	53.75 ± 1.86	C	64.11 ± 5.16	49.80 ± 4.79	B
16	48.76 ± 7.39	32.73 ± 3.64	A	65.78 ± 1.86	54.93 ± 1.57	C	63.13 ± 0.51	51.01 ± 0.36	C
17	50.64 ± 0.70	31.18 ± 5.41	B	65.53 ± 2.59	56.47 ± 0.16	C	64.53 ± 2.33	51.62 ± 1.15	C

**Table 5 foods-14-00889-t005:** Condition and result of central composite design.

Run No.	Independent Variables ^1^			Alaska Pollock ^2^				Golden Threadfin Bream ^2^				Hairtail ^2^			
	*x* _1_	*x* _2_	*x* _3_	*y* _1_	*y* _2_	*y* _3_	*y* _4_	*y* _1_	*y* _2_	*y* _3_	*y* _4_	*y* _1_	*y* _2_	*y* _3_	*y* _4_
				g/cm^2^	g	Pa	Pa·s	g/cm^2^	g	Pa	Pa·s	g/cm^2^	g	Pa	Pa·s
1	45	85	10	391.2	−273.4	19,096.8	20,184.7	148.5	−139.3	7401.1	8458.1	147.4	−132.8	3245.9	3546.4
2	155	85	10	415.5	−297.8	25,157.0	25,802.3	165.1	−157.3	9066.5	10,630.8	148.4	−115.8	4961.1	5365.1
3	45	315	10	211.9	−183.3	10,636.4	11,067.3	106.9	−90.0	4242.8	4829.6	62.4	−52.3	1701.9	1778.2
4	155	315	10	227.9	−190.0	12,855.5	13,328.3	102.1	−93.0	5273.1	6291.3	63.8	−49.6	1889.3	2068.0
5	45	85	40	332.6	−259.7	15,008.8	15,647.1	54.6	−41.3	2041.2	2562.8	75.3	−62.3	3237.5	4001.3
6	155	85	40	343.1	−268.1	23,972.5	25,489.0	65.0	−46.2	5166.3	5704.4	113.5	−97.0	6226.6	7220.4
7	45	315	40	208.3	−171.0	8839.9	9266.0	45.3	−35.4	1390.4	1741.9	68.2	−63.6	2784.4	3330.3
8	155	315	40	228.0	−186.4	13,764.7	14,660.7	51.5	−38.0	2005.0	2526.6	67.4	−64.3	3232.8	4042.5
9	10	200	25	287.5	−203.3	11,306.9	11,711.5	63.1	−52.5	3131.1	3769.5	99.7	−93.3	3845.6	4520.2
10	190	200	25	306.8	−235.7	18,504.1	20,035.5	79.6	−71.5	8378.2	10,371.9	105.0	−97.0	6045.2	7115.3
11	100	10	25	509.2	−334.8	24,868.1	26,186.8	100.3	−88.0	6691.0	7844.8	150.9	−131.5	8570.0	10,389.5
12	100	390	25	198.8	−153.4	9122.1	9530.2	51.7	−47.5	2611.5	3487.2	68.5	−62.3	3289.6	3873.4
13	100	200	1	264.6	−225.8	16,766.7	17,330.0	179.2	−171.6	12,323.5	13,300.1	120.0	−102.5	9884.9	10,310.2
14	100	200	50	257.0	−210.6	12,572.6	13,557.7	40.8	−28.6	1496.1	1821.0	64.3	−55.3	2813.7	3292.8
15	100	200	25	271.8	−213.3	13,500.0	14,178.1	67.8	−59.2	3783.6	4601.5	98.2	−94.3	4784.0	5668.1
16	100	200	25	283.5	−210.0	14,045.3	14,688.8	65.5	−56.5	4752.3	5834.1	99.4	−90.1	4772.4	5636.2
17	100	200	25	285.6	−215.9	14,068.0	14,433.4	67.4	−57.0	4731.4	5604.5	102.6	−97.3	5151.9	6062.0

^1^ *x*_1_: starch content; *x*_2_: water content; *x*_3_: salt content; ^2^ *y*_1_: hardness; *y*_2_: adhesiveness; *y*_3_: storage modulus (G′); and *y*_4_: complex viscosity.

**Table 7 foods-14-00889-t007:** Verification of optimal hardness and adhesiveness.

Sample	Starch (*x*_1_)	Water (*x*_2_)	Salt (*x*_3_)	Run	Hardness	Adhesiveness
					g/cm^2^	g
Alaska pollock	153.6	79.1	28.7	Predicted value	398.4	−286.7
				Actual value	385.3	−294.8
				*p*-value	0.425	0.284
	91.8	182.7	10.8	Predicted value	296.7	−227.7
				Actual value	313.7	−220.6
				*p*-value	1.000	0.997
Golden threadfin bream	190	10	14	Predicted value	176.7	−166.7
				Actual value	134.1	−121.3
				*p*-value	0.605	0.267
	190	340.5	1.57	Predicted value	146.2	−139.7
				Actual value	143.3	−123.3
				*p*-value	0.111	0.158
Hairtail	164.5	60	16.8	Predicted value	158.2	−130.4
				Actual value	173.1	−140.8
				*p*-value	0.814	0.274
	149.9	82.7	13.8	Predicted value	152.2	−127.3
				Actual value	160.2	−135.0
				*p*-value	0.990	0.487

**Table 6 foods-14-00889-t006:** Coefficients of the regression model for sample and physical properties, and the results of the significance tests.

Physical Properties	Model ^1^	Alaska Pollock			Golden Threadfin Bream			Hairtail		
		Coefficient	*t*-Value	*p*-Value	Coefficient	*t*-Value	*p*-Value	Coefficient	*t*-Value	*p*-Value
Hardness (*y*_1_)	*R* ^2^	96.48%			99.37%			96.48%		
	Intercept	530.30	11.33	0.00	244.50	18.85	0.00	190.90	8.56	0.00
	*x* _1_	−0.0560	−0.12	0.91	−0.0130	−0.09	0.93	0.1690	0.74	0.48
	*x* _2_	−1.6900	−7.54	0.00	−0.3597	−5.79	0.00	−0.4110	−3.85	0.01
	*x* _3_	−0.2400	−0.14	0.89	−7.8410	−16.47	0.00	−2.0430	−2.50	0.04
	*x* _1_ ^2^	0.0012	0.61	0.56	0.0009	1.64	0.15	−0.0004	−0.47	0.66
	*x* _2_ ^2^	0.0018	4.33	0.00	0.0003	2.72	0.03	0.0001	0.56	0.59
	*x* _3_ ^2^	−0.0434	−1.71	0.13	0.0778	11.07	0.00	−0.0222	−1.83	0.11
	*x* _1_ *x* _2_	0.0000	0.02	0.99	−0.0005	−1.77	0.12	−0.0008	−1.56	0.16
	*x* _1_ *x* _3_	−0.0015	−0.19	0.85	0.0007	0.34	0.75	0.0053	1.41	0.20
	*x* _2_ *x* _3_	0.0092	2.44	0.05	0.0059	5.65	0.00	0.0084	4.68	0.00
Adhesiveness (*y*_2_)	*R* ^2^	98.90%			99.45%			93.40%		
	Intercept	−346.00	−18.05	0.00	−232.00	−18.74	0.00	−168.10	−6.39	0.00
	*x* _1_	−0.0600	−0.31	0.77	−0.0820	−0.65	0.54	−0.0390	−0.14	0.89
	*x* _2_	0.8038	8.75	0.00	0.3887	6.55	0.00	0.3170	2.51	0.04
	*x* _3_	1.1610	1.65	0.14	7.7020	16.93	0.00	1.3760	1.42	0.20
	*x* _1_ ^2^	−0.0008	−1.02	0.34	−0.0006	−1.19	0.27	0.0007	0.66	0.53
	*x* _2_ ^2^	−0.0009	−4.94	0.00	−0.0003	−2.60	0.04	0.0001	0.44	0.67
	*x* _3_ ^2^	−0.0090	−0.86	0.42	−0.0734	−10.93	0.00	0.0364	2.55	0.04
	*x* _1_ *x* _2_	0.0002	0.50	0.63	0.0003	1.26	0.25	0.0004	0.67	0.53
	*x* _1_ *x* _3_	0.0011	0.34	0.74	0.0020	0.97	0.36	−0.0084	−1.88	0.10
	*x* _2_ *x* _3_	−0.0020	−1.29	0.24	−0.0072	−7.21	0.00	−0.0084	−3.93	0.01
G′ (*y*_3_)	*R* ^2^	99.33%			89.31%			48.08%		
	Intercept	26,522.00	16.16	0.00	10,711.00	2.81	0.03	3598.00	0.58	0.58
	*x* _1_	23.6000	1.40	0.21	19.8000	0.50	0.63	63.3000	0.99	0.36
	*x* _2_	−71.9000	−9.15	0.00	−3.0000	−0.16	0.88	1.9000	0.06	0.95
	*x* _3_	−310.3000	−5.15	0.00	−362.0000	−2.58	0.04	−54.0000	−0.24	0.82
	*x* _1_ ^2^	0.1662	2.52	0.04	0.0470	0.31	0.77	−0.2030	−0.81	0.44
	*x* _2_ ^2^	0.0958	6.44	0.00	−0.0199	−0.58	0.58	−0.0172	−0.31	0.77
	*x* _3_ ^2^	1.8650	2.10	0.07	2.6300	1.27	0.25	−0.5200	−0.15	0.88
	*x* _1_ *x* _2_	−0.1577	−4.36	0.00	−0.0629	−0.75	0.48	−0.0810	−0.59	0.57
	*x* _1_ *x* _3_	0.8610	3.10	0.02	0.1600	0.25	0.81	0.2400	0.22	0.83
	*x* _2_ *x* _3_	0.3220	2.43	0.05	0.2300	0.75	0.48	0.0860	0.17	0.87
Complex viscosity (*y*_4_)	*R* ^2^	99.59%			89.53%			47.83%		
	Intercept	28,043.00	20.98	0.00	11,533.00	2.79	0.03	3645.00	0.53	0.61
	*x* _1_	12.40	0.90	0.40	20.80	0.49	0.64	68.90	0.97	0.36
	*x* _2_	−75.53	−11.80	0.00	−5.00	−0.25	0.81	−1.30	−0.04	0.97
	*x* _3_	−335.00	−6.83	0.00	−339.00	−2.24	0.06	9.00	0.04	0.97
	*x* _1_ ^2^	0.20	3.77	0.01	0.08	0.46	0.66	−0.23	−0.83	0.43
	*x* _2_ ^2^	0.10	8.29	0.00	−0.02	−0.57	0.59	−0.01	−0.23	0.83
	*x* _3_ ^2^	2.04	2.81	0.03	1.90	0.85	0.42	−1.63	−0.44	0.68
	*x* _1_ *x* _2_	−0.15	−5.24	0.00	−0.06	−0.67	0.53	−0.08	−0.53	0.61
	*x* _1_ *x* _3_	1.12	4.94	0.00	0.04	0.06	0.95	0.28	0.24	0.82
	*x* _2_ *x* _3_	0.32	2.94	0.02	0.29	0.86	0.42	0.09	0.16	0.88

^1^ *x*_1_: starch content; *x*_2_: water content; and *x*_3_: salt content.

## Data Availability

The original contributions presented in this study are included in this article. Further inquiries can be directed to the corresponding author.

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
