# Peer review of "Optimizing 3D Food Printing of Surimi via Regression Analysis: Physical Properties and Additive Formulations"

_foods, 2025, doi:10.3390/foods14050889_

Round 1

Reviewer 1 Report

Comments and Suggestions for Authors

The manuscript aims to establish a universal range of additive contents for the 3D printing of food-based inks based on the physical properties of three types of surimi: Alaska pollock, golden threadfin bream, and hairtail. The findings will contribute to increasing the efficiency of 3D food printing and enhancing the development of foods tailored to consumer preferences. Overall, this research is interesting and presented many valuable results.

I have listed some specific questions below.

(1) The physical properties of ink mainly include hardness, elasticity, viscosity, chewability, cohesiveness, resilience. Why did the authors just choose only hardness and cohesiveness as the indicators for the surimi ink?

(2) The most critical aspect of 3D printing is the quality of the printed output. However, as illustrated in Figure 1, the quality of the printed products is generally subpar. Groups A and B exhibit numerous burrs, while group C shows signs of deformation, indicating that the performance of the surimi ink is inadequate. The author's subsequent analysis relies on the characteristics of surimi ink, which demonstrates poor printing effects, rendering all final analysis results lacking in practical guidance.

(3) To articulate the rheological properties of ink, the following articles are recommended.

ü  “Effect of various physical modifications of pea protein isolate (PPI) on 3D printing behavior and dysphagia properties of strawberry-PPI gels”.

ü  “Pea protein-xanthan gum interaction driving the development of 3D printed dysphagia diet”.

(4) Line 98 Why was the food processor maintained under vacuum ?

(5) Figure 3 lacks clarity.

Author Response

Thank you for considering our article for publication in Foods.

We are grateful for the valuable suggestions provided. Here are our responses to the comments of Reviewer

Reviewer 2 Report

Comments and Suggestions for Authors

The authors report a study of optimization of a surimi-based food ink for 3D printing. The topic is innovative in the food industry and is interesting, but the paper has numerous critical elements.

INTRODUCTION

The structure of the introduction needs some revision. There is some fluidity up to line 50, but then this stops and suddenly arrives at the surimi. The organization of the paragraphs should therefore be revised.

- Line 47: which is the "trade-off relationship between plasticity/fluidity and cohesiveness when adding hydrocolloids and lipid binders"?

From the state of the art it is not clear if a surimi food ink has been ever developped.

- what you mean by "developing a universal method" ? how can you manage to develop one single procedure regardless of the fish species? 

- I suggest to improve the aim section, it is very "dispersive" without going straight to the point.

- The printability procedure is not referenced? is your method?

- 2.6: is not a "Central composition" but Central Composite. Moreover, in this paragraph the structure of the CCD is completely missing, this must be described. Furthermore, an explained rationale for the selection of the ingredient and their range in the CCD must be disclosed.

- Why the printability was not choose as response in the CCD model? this is a mistake in my opinion, as these experimental runs don't consider the printer precision which should be correlated to the rheological and mechanical properties.

- From the Table 3 i see that you evaluated the printability of the different experimental runs, so why don't include them in the model? in that way it was easly observable the interactions between the indipendent variables.

- from figure 2 it is not clear to me how these correlations were made, since there are 17 different formulations for each surimi, which values ​​were correlated? more explanations should also be given in the caption.

- The figure 3B also is not clear, usually in CCD a surface plot is reported to show these types of interactions.

- Even from figure 3 it is not clear to me how this "optimal range" was established. Furthermore, I believe that the range is definitely too wide for this kind of optimization. Typically an optimal point is obtained from polynomial regression, from which an optimal contour or area can be extrapolated (expressed via contour plot). But in this way the concept of "optimal" seems too generic and broad to me.

- Figure 5 is incomprehensible to me. From Minitab you can generate better graphics to show the interactions between indipendent factors.

- All the figure captions are too narrows, these must be improved.

I suggest some strong modification to the paper before resubmitting.

Author Response

(The authors gave the same response as above.)

Reviewer 3 Report

Comments and Suggestions for Authors

Overall, the paper lacks sufficient details in the methods and discussion sections, which diminishes its scientific rigor and the value of its conclusions.

Line 24: The statement is overly generic. Please specify the types of rheological properties studied and express the key results clearly.

Methodology

Line 91: The section lacks detailed information on the preparation of surimi. Please provide more specifics about the materials and procedures used.

For the Central Composite Design (CCD), include a detailed description of the variables and their levels, as well as the total number of experiments recommended by the CCD.

Discussion

Figure 2: The graph showing the correlation coefficient has identical x- and y-axis labels. Please revise if necessary to ensure clarity and proper representation of the data.

Figure 3: The captions for Figures 3a and 3b are not clearly specified. Please revise to clearly identify what each sub-figure represents.

The discussion section is weak and lacks depth. It provides limited insights into the significance of the results. Please strengthen this section by elaborating on the implications of your findings.

Conclusion

Include the optimal conditions obtained from your experiments in the conclusion section to enhance clarity and provide practical value to your findings.

Author Response

(The authors gave the same response as above.)

Round 2

Reviewer 1 Report

Comments and Suggestions for Authors

I have read the responses from the authors for the manuscript (Manuscript ID: foods-3451863). I suggest it can be accepted in the current state.

Author Response

I sincerely appreciate your time and effort in reviewing my manuscript.

Lee

Reviewer 3 Report

Comments and Suggestions for Authors

The authors have revised the article, and I believe that the updated version is suitable for publication.

Author Response

(The authors gave the same response as above.)
